# The Detection of Substitution Adulteration of Paprika with Spent Paprika by the Application of Molecular Spectroscopy Tools

**DOI:** 10.3390/foods9070944

**Published:** 2020-07-16

**Authors:** Pamela Galvin-King, Simon A. Haughey, Christopher T. Elliott

**Affiliations:** ASSET Technology Centre, Institute for Global Food Security, Queen’s University Belfast, 19 Chlorine Gardens, Belfast BT9 5DL, Northern Ireland, UK; p.galvin-king@qub.ac.uk (P.G.-K.); chris.elliott@qub.ac.uk (C.T.E.)

**Keywords:** paprika, near-infrared, Fourier transform infrared, economically motivated adulteration, chemometrics

## Abstract

The spice paprika (*Capsicum annuum* and *frutescens*) is used in a wide variety of cooking methods as well as seasonings and sauces. The oil, paprika oleoresin, is a valuable product; however, once removed from paprika, the remaining spent product can be used to adulterate paprika. Near-infrared (NIR) and Fourier transform infrared (FTIR) were the platforms selected for the development of methods to detect paprika adulteration in conjunction with chemometrics. Orthogonal partial least squares discriminant analysis (OPLS-DA), a supervised technique, was used to develop the chemometric models, and the measurement of fit (R^2^) and measurement of prediction (Q^2^) values were 0.853 and 0.819, respectively, for the NIR method and 0.943 and 0.898 respectively for the FTIR method. An external validation set was tested against the model, and a receiver operating curve (ROC) was created. The area under the curve (AUC) for both methods was highly accurate at 0.951 (NIR) and 0.907 (FTIR). The levels of adulteration with 100% correct classification were 50–90% (NIR) and 40–90% (FTIR). Sudan I dye is a commonly used adulterant in paprika; however, in this study it was found that this dye had no effect on the outcome of the result for spent material adulteration.

## 1. Introduction

Paprika is a spice best known for its use in a wide variety of cooking methods for both flavour and colour. It can be found in a variety of foods, including seasonings and sauces [1]. The European Spice Association [2] lists *Capsicum annuum* and *frutescens* species of paprika, which is a member of the family Solanaceae. Paprika consists of dried ground peppers of the sweet and slightly pungent varieties, depending on its origin and grade. The characteristic reddish colour is due to the presence of carotenoids [1].

Spices such as paprika are often the target for food fraud as they are valuable commodities, and fraudsters aim to deceive consumers into thinking they are buying authentic and safe spices [3]. Paprika is also commonly found in many processed foods, and therefore, any fraud in this spice may pose a huge risk to consumer safety [4]. Different forms of adulteration in paprika has been found to include substitution adulteration with waste or inferior products, falsely declared origin [5] and addition adulteration with the use of illegal dyes such as the commonly found Sudan I and IV according to the Rapid Alert System for Food and Feed (RASFF) portal (https://ec.europa.eu/food/safety/rasff/portal_en).

Examples of substitution adulteration with waste products include the adulteration by bulking with white pepper, curcuma, brick powder and barium sulphate [6]. Adulteration in paprika also involved the addition of the nut protein from almonds in place of paprika [7]. This case indicated the carelessness of the criminals and the serious public health threat that can arise from adulteration, even when only economic gain is the motivation as anaphylaxis can occur in susceptible individuals. 

Falsely declared origin has also been an issue with paprika adulteration. In 2004, Hungarian paprika was found to be incorrectly marketed as ‘domestic Hungarian samples’ when the aflatoxin that was found was from a fungus that could not have originated in Hungary due to climate. It therefore became obvious that fraud was occurring. The Hungarian paprika had been mixed with paprika from South America following a drought in the summer previously [8]. The threat of adulteration means the authentication of paprika from the Murcia and La Vera region of western Spain is important as they have protected designation of origin (PDO) status [5,9,10,11]. The authentication of Szegedi paprika from Hungary as another product with PDO status is also essential [12]. Various methods have been developed to identify these protected spices and characterize the PDO status, including DNA typing methods [11], free zone capillary electrophoresis (FZCE) [5], elemental analysis along with chemometrics [10,12] and UV-Vis with chemometrics [9]. Characteristic fingerprints based on phenolic compounds of paprika have also been obtained from chromatographic approaches, with high-performance liquid chromatography ultraviolet (HPLC-UV) and HPLC-electrochemical detection (EC) being used alongside chemometrics to determine varieties and origin of paprika [13,14].

To enhance the colour and value, dyes may also be used to adulterate paprika. In 1994, lead oxide was added to paprika to enhance colour, which resulted in the hospitalisation of many consumers [15]. Dyes found in paprika include Sudan I, Sudan IV, E160b, Orange II, Rhodamine B and Para Red [16], with Sudan I and IV being the most commonly found, which are possibly carcinogenic and potentially genotoxic [17].

Spectroscopy and chemometrics are increasingly becoming the chosen methods for adulteration detection in herbs and spices. It is a preferable form of analysis for the detection of adulteration as it offers a robust, rapid and inexpensive form of analysis which requires little expertise to carry out analysis once the test method and chemometrics are in place. Chemometrics is used to extract the relevant information from the spectra obtained. Used alongside spectroscopy, it is a powerful tool to allow for the classification of adulterated and authentic products with successful applications, including the detection of adulteration of garlic, ginger, oregano and onion powder [3]. In Table 1, a number of spectroscopic techniques used alongside chemometrics for the detection of various forms of adulteration of paprika have been outlined.

In the literature, most studies on the adulteration of paprika using spectroscopy involve the detection of dyes. There have been a small number of investigations into detection techniques for bulking agents. In a study by Horn et al. [4], results indicated >80% sensitivity and specificity when Fourier transform infrared (FTIR) was used to detect lead oxide (3%), lead chromate (3%), silicon dioxide (5%), poly vinyl chloride (10%) and gum Arabic (10%), and, in addition, Sudan I and IV were detected to 1%. Galaxy Scientific used classical least-squares (CLS)-based Advanced-ID algorithm to detect bulking agents tomato skin (0.5%) and brick dust (5%) [18]. Oliveira et al. [19] used portable near-infrared (NIR) spectroscopy to detect potato starch, acacia gum and annatto, and it was found to be capable of detecting adulteration of paprika both qualitatively using partial least squares- discriminant analysis (PLS-DA) and quantitatively using partial least squares regression (PLSR). The PLS-DA models showed a specificity >90% and lower than 2% error. The R^2^ and root mean square error of prediction (RMSEP) values for the PLSR were 0.95 and 2.12 (potato starch), 0.97 and 1.68 (acacia gum), and 0.87 and 1.74 (annatto). The use of portable spectroscopy in this way can be highly valuable at detecting adulteration at various points along the supply chains, and for this reason, it can also act as a major deterrent to fraudsters.

The detection of dyes in paprika has been undertaken using a number of spectroscopic techniques including Fourier transform near-infrared (FT-NIR), FTIR, Raman, nuclear magnetic resonance (NMR), synchronous fluorescence spectroscopy (SFS) and UV-vis. These methods detected mainly Sudan I dye; however, methods were also developed to detect Sudan II, III, IV and Congo Red and annatto. The chemometrics used involved the detection of dyes by both qualitative and quantitative methods (Table 1). The addition of dyes improves colour and subsequently may add value to the product. The paler reds and brown shades of paprika are the poorest quality and are also the most pungent [1].

According to the herb and spice industry, a bulking agent used in substitution adulteration is spent paprika. The ESA describes a spent material as one that has ‘…any valuable constituent omitted or removed which misleads the customer (e.g., spent and partially spent spices and herbs, de-oiled material, defatted material)’ [32]. Paprika oleoresin is extracted from the fruit and is an oil-soluble extract. It is well known for its colouring properties and can be found in cheese, orange juice, sweets and sauces. Paprika oleoresin, valued in the European market at €126 million in 2015, accounts for 25% of the overall oleoresin market globally [33]. Once this oleoresin is removed from paprika, the remaining ‘spent’ material is then a waste product. The current method (American spice trade association (ASTA) method 26.1) for the detection of defatted paprika in paprika involves the detection of a colour change reaction by microscopy following the addition of a sulphuric acid and boric acid reagent [34]. This method, however, requires highly trained personnel. There do not appear to be any other methods reported on in literature for the detection of spent paprika in paprika other than testing to determine if the overall quality standards are met in the ESA Quality Minima Document [35,36]. Once this spent material is used as a substitute for paprika, the colour of the product is less vibrant. There is therefore a risk that a dye may be used alongside the spent material to ensure appearances are upheld.

The aim of this study was to develop a rapid and robust screening technique to detect economically motivated adulteration in paprika with spent material using the spectroscopic platforms NIR and FTIR in conjunction with chemometrics. Authentic paprika and spent paprika samples were collected and analysed by spectroscopy to create a database of representative samples for the classes ‘Paprika’ and ‘Adulterant’ in the chemometric models. External validation was carried out to determine the correct classification rate of the models and their potential to determine adulteration with spent material in unknown paprika samples. Sudan I dye was also added to adulterated samples to determine is this affected the ability of the method to detect adulteration with spent paprika.

## 2. Materials and Methods

### 2.1. Sample Collection

A total of 159 samples were collected for the development of a chemometric model to detect spent material in paprika samples. The paprika (*n* = 140) and spent paprika samples (*n* = 19) came in powdered form. The samples were provided by highly reliable sources from leading herb and spice industry suppliers. The paprika samples originated from Peru, China, Hungary and Spain with spent material originating from China, India and Spain. Paprika samples also included those which were processed with stone milling and hammer milling. The samples had a range of American spice trade association (ASTA) colour values (extractable colour of paprika) from 75 to 269. Mixtures of varying seed/pod ratios were also obtained. The extraction procedures for the spent material included the use of hexane and acetone/hexane extraction solvents.

### 2.2. Preparation of Samples

The samples were milled prior to receipt in the laboratory. For NIR analysis, no further sample preparation was carried out. Prior to FTIR analysis, the samples were milled further to improve homogeneity of the samples for the small sample testing area of 1.8 mm on the diamond crystal of the attenuated total reflectance (ATR) accessory. Approximately 10 g of each sample was added to the grinding jars of a Planetary Ball Mill PM 100 (Retsch, Haan, Germany) and milled at 500 rpm for 5 min.

### 2.3. NIR Analysis

The paprika and spent paprika samples were analysed on the Thermo Antaris II FT-NIR (Thermo Fisher Scientific, Dublin, Ireland). Data were collected in reflectance mode, with spectral data output measured in absorbance units. Approximately 10 g of each sample was placed into a sample cup (minimum of 0.5 cm depth) for analysis, and the samples were run on the integrating sphere module of the instrument. Prior to each analysis, a background scan was performed. The spectral data were then collected from a rotating sample with a resolution of 8 cm^−1^ in the range of 4000–12,000 cm^−1^. The samples were analysed in triplicate and remixed prior to each spectral data collection. A total of 64 scans were acquired for each of the spectra.

### 2.4. FTIR Analysis

Mid-infrared spectral data were collected on the Thermo Nicolet iS5 FTIR (Thermo Fisher Scientific, Dublin, Ireland) with diamond crystal on the ATR accessory, ZnSe lens and DTGS KBr detector. Following milling, the samples were placed onto the diamond crystal sampling area of the ATR accessory, and the slip clutch pressure tower was lowered into position. This improves reproducibility between samples as it ensures equal pressure is applied to the samples prior to analysis. A total of 32 scans were acquired for each of the spectra, and the spectral data ranged from 550 to 4000 cm^−1^ at 4 cm^−1^ resolution. All samples were analysed in triplicate and averaged prior to chemometric model development. Further sample collection parameters included: 47 seconds collection length, 0 levels of zero filling, N-B strong apodization, mertz phase correction, 11,742.96 cm^−1^ laser frequency, 12,415 scan points and a background gain of 4.00.

### 2.5. Chemometrics

The development of chemometric models was undertaken using SIMCA 15 (Sartorius, Sweden). The qualitative models created in SIMCA involved firstly pre-processing the raw data from the FTIR and NIR. This involved the use of Standard Normal Variate (SNV), 1st/2nd Derivative, Savitzky Golay (SG) with 15 points and a quadratic polynomial order along with Pareto scaling. Pre-processing prior to model development allowed focus on the important data points [37]. PCA, an unsupervised technique, was performed initially, to determine if separate classes could be observed based on the spectral data from both NIR and FTIR for paprika and spent paprika. Following this, a supervised orthogonal partial least squares discriminant analysis (OPLS-DA) model was created to further improve the qualitative models for both NIR and FTIR spectral data. The OPLS-DA algorithm uses both predictive (correlated) and orthogonal (uncorrelated) components to create the classification model to offer a greater understanding of all the aspects of the data. Chemometric analysis was carried out in the range of 550–1800 cm^−1^ and 2800–4000 cm^−1^ for FTIR analysis and 4000–9000 cm^−1^ for NIR analysis. The classes for the binary chemometric models were made up of ‘Paprika’ (*n* = 104) and ‘Adulterant’ (*n* = 17).

### 2.6. Validation Procedure

The validation procedure for the NIR and FTIR paprika adulteration models was based on recommendations from the ‘Guidance on Validating Non-Targeted Methods for Adulteration Detection’ [38], Riedl et al. [39] and McGrath et al. [40].

#### 2.6.1. Internal Cross-Validation

The software SIMCA 15 carried out internal cross-validation of the chemometric models. The averaged data were divided into 7 parts, and each 1/7th was removed in turn. Each time, a new model was created using the 6/7th of the data. The 1/7th that had been removed was then predicted using the new model and compared to the original data. From this, the Predicted Residual Sum of Squares (PRESS) was calculated. PRESS was converted into Q^2^ by dividing by the sum of squares and subtracting from one. This was used as an indicator of the predictability of the model. The explained variation of the real data from the model is represented by the R^2^ value, measurement of fit. The closer both R^2^ and Q^2^ are to 1, the better the model. These values determined which models would be used for external validation.

#### 2.6.2. External Validation

All samples chosen for external validation were removed from the chemometric model set. The external validation set was made up of authentic (typical) paprika samples (*n* = 30) and spiked samples (atypical) (*n* = 90). To carry out the spiking, two spent paprika were used. Each spent paprika was used to spike five authentic paprika samples at 10–90% levels, therefore resulting in 90 spiked samples in total. The five authentic paprika samples chosen for spiking were selected out of a range of six samples. The samples used for external validation included a range of ASTA levels, milling techniques and countries of origin. As suggested in the guidance from US Pharmacopoeia [38], the external test samples were chosen from the model centroid; otherwise, the external validation samples may portray the method inaccurately.

A binary model was created for both the NIR and FTIR data as the aim of this work was to focus on the detection of spent paprika in paprika. The external test set of authentic 100% paprika samples (typical) and 90 spiked paprika samples (atypical) were run against the chosen OPLS-DA model. Following this, a receiver operating curve (ROC) curve was developed to plot the true positive rate (TPR) against the false positive rate (FPR) to determine the performance of the method.

### 2.7. Sudan Dye Detection in Spent Material

Sudan I was added to 100% spent and 50% spent samples at levels of 0.1%, 0.5%, 1%, 2.5% and 5%. These samples were then analysed on the FTIR and NIR instruments according to the aforementioned procedures. They were then tested as unknowns against the chosen chemometric models to determine if the addition of Sudan dye affected the model’s ability to detect spent material.

## 3. Results and Discussion

### 3.1. Raw Spectral Data

The spectral data for paprika and spent paprika can be seen in Figure 1. The visual differences between the spectra are circled in the images below.

The NIR spectra of 100% paprika and spent material show that there is a clear distinction between the bands that correspond to oil, C-H bonds, (4100–4400 cm^−1^, 5350–6000 cm^−1^) and the water band O-H (5000–5200 cm^−1^). The contrast in oil is expected as the spent material will have oils extracted from it in the form of oleoresin. As NIR contains overtones and combinations of fundamental vibrations, the specific bands are weaker in intensity and visually more difficult to distinguish, hence the value of chemometrics in extracting information.

The differences in the functional group region of the FTIR spectra occur at the C-H region (2700–3000 cm^−1^) and the O-H region (3000–3600 cm^−1^). There are many variations in the bands of the fingerprint region between the spent material and paprika. The most obvious band differences between paprika and spent material occur at 1743 cm^−1^ (C=O) [4] where bands in the paprika samples can be seen, but not in the spent paprika. These bands can be considered diagnostic tools and enhance the possibility to distinguish authentic from adulterated paprika. Although the spectra visually show differences at 100% levels when paprika and spent paprika spectra are compared, the level of adulteration can vary in adulterated samples, and may not be so clear without the use of chemometric software to extract further information from the raw spectral data. From both the NIR and FTIR spectra, differences between the spent material and paprika bands can be observed here, indicating the potential of the methods using spectroscopy in conjunction with chemometrics.

### 3.2. Chemometric Models

Following the collection of raw data, chemometric models were created using the software SIMCA 15 for both NIR and FTIR. The R^2^ and Q^2^ values for the models were calculated in the software and used to determine which models performed the best. These models can be seen in Figure 2. 

In Figure 2, the NIR classification models for paprika and spent paprika were developed using the unsupervised technique PCA and the supervised technique OPLS-DA. These algorithms were carried out following the pre-processing, SNV, 1st derivative (PCA) 2nd derivative (OPLS-DA), SG and Pareto scaling. In the PCA model, the first four principal components showed 93.6% variation. Separation could be seen between the spent paprika and the paprika in the unsupervised model indicating model reliability as the principal components indicate the maximum variance between the spent material and paprika. This showed that the separation in the model was reliable. The supervised classification model OPLS-DA was then created and proved to have a good R^2^ (0.853) and Q^2^ (0.819) value. The OPLS-DA supervised technique improves the separation of classes with the use of predictive (correlated) and orthogonal (uncorrelated) components.

In Figure 3, the classification models of spent material and paprika shown were developed using PCA and OPLS-DA algorithms with the spectral data from FTIR. The principal components showed 80.6% variation in the first four components. The pre-processing of these models involved SNV, 1st derivative, SG and Pareto scaling. The OPLS-DA model produced an R^2^ value of 0.943 and a Q^2^ value of 0.898.

### 3.3. External Validation Results 

Following the raw data collection of the spent and paprika samples, the authentic paprika samples were randomized so as to prevent a group of similar products from similar suppliers being grouped together. They were then split into a training set for use in the model development (*n* = 104) and test set for use in the validation set (*n* = 30). This allowed a separate set of authentic samples to be used for external validation of the methods, similarly to the procedure explained by Riedl et al. [39]. A further set of spiked samples were also used for the external validation set (*n* = 90) as outlined in the External Validation section. The training set was used to develop and optimise the best model based on the R^2^ and Q^2^ performance results as explained in the Internal Validation section. As advised by the US Pharmacopoeia, ROC curves were created for the NIR and FTIR data sets following the external test set prediction to determine the performance of the chosen models and can be seen in Figure 4.

The TPR and the FPR were plotted against each other to create the ROC curve to determine model performance for both NIR and FTIR data. The TPR and FPR, referred to by SIMCA, are the sensitivity rate of the test (TPR), and 1-specificity (FPR) (Figure 4). In this study, the sensitivity refers to the rate of correctly identified unadulterated paprika samples and the specificity refers to the rate of correctly identified adulterated samples.

The area under the curve (AUC) of the ROC curve indicates method performance following the testing of external samples. The diagonal line indicates an AUC of 0.5, and this portrays the result of a random decision. The closer the AUC is to 1, the better the model performance [38]. The AUC for the NIR dataset and the FTIR dataset was 0.951, and 0.907 respectively. According to the guidelines, an AUC greater than 0.9 is considered highly accurate [41] based on recommendations by Swets et al. [42].

The AUC gives an indication of model performance; however, there is still a requirement to calculate the test method’s cut-off point. This cut-off point is a predictive score value that indicates the optimal cut-off point for correct classification. It is required as each sample tested produces a predicted score value, and the cut-off point determines whether a sample is adulterated or not by a comparison of its predictive score value. This optimal cut-off point is determined by the calculation of the Youden index (Youden, 1950).

The Youden index (J) is calculated to choose the optimal threshold for the test method using the calculation J = Sensitivity + Specificity − 1. The Youden index not only provides an optimal cut-off point for a diagnostic test, but it also facilitates a comparison between tests [41,43]. As SIMCA plotted the TPR and FPR for the ROC plot, the Youden index was calculated through these values as J = TPR-FPR.

The Youden index ranges from 0 to 1 with 1 being the best possible outcome (Youden, 1950). The Youden index for NIR was calculated as 0.788, and for FTIR it was 0.733. Once the Youden index was calculated, the corresponding cut-off value for the test was identified from the predicted score value on the classification list from SIMCA 15. This cut-off value was 0.737 for NIR and 0.922 for FTIR. Therefore, any unknown sample being predicted using this model has its predicted score value compared to the cut-off for the test. A sample ≥ the cut-off is considered paprika, but a sample with a value < the cut-off value is determined as adulterated. The results of the validation test according to the test cut-off calculations set can be seen in Table 2.

According to the ASTA analytical method 26.1 for the detection of defatted material in paprika, it is expected that adulteration of spent paprika would be at high concentrations, above 20% [34]. As observed in Table 2, at the cut-off point determined by the Youden index, the external validation results for the detection of spent paprika in paprika indicate that even at 40% (NIR) and 30% (FTIR) adulteration levels, this method did not detect all ten samples with adulteration at these levels. The NIR was more accurate at detecting 100% paprika samples; however, it also had a reduced ability to detect the adulterated samples. Conversely, the FTIR method was slightly more accurate with the adulteration detection (100% at 40% adulteration level) and less accurate with the correct classification of the 100% paprika samples (83.3%). The results indicated in Table 2 are calculated based in the cut-off calculated by the Youden index, assuming that false positives and false negatives are weighted equally.

The difficulties with this method may be as a result of the fact that both the spent material and paprika are from the same part of the same plant. The level of oleoresin removed from the spent material is also unknown, and this could affect the outcome of the result. It has also been reported that Bate’s method cannot detect all forms of spent material, and therefore, investigation into other techniques was required [44]. Although some difficulties can be seen with spectroscopy, it does have some prospects as a first port of call for adulteration as a screening technique. The separation seen in the chemometric models indicates that clear differences are observed, but the external validation results indicate that those difference are just not so clear-cut at the lower percentage levels. This method does have benefits above some others such as DNA techniques, which could not be used due to the fact the spent material is part of the same plant.

Spent material has been detected in black pepper using non targeted methods at 10–30% [45] and ≥20% by Lafeuille et al. [44]. Spent black pepper can be derived from light berries as they are used for oil extraction [44]. Light berries are berries without a seed/kernel [46] and are therefore different in nature from a typical black peppercorn. Even prior to oil extraction, they are likely to produce different spectroscopic results to black peppercorns. In contrast, according to the Commission Regulation (EU) No 231/2012 for food additives, spent paprika is produced from ground fruit pods [47], a similar part of the plant as paprika spice. Therefore, the spent material from light berries may be more easily detected than spent material from paprika.

A combination of detection methods is required to verify results in a two-platform approach. Both microscopy techniques and gas chromatography mass spectrometry (GC-MS) were used to detect adulteration in fennel seeds [48]. Garber et al. [49] illustrated the need for a range of analytical techniques when mass spectrometry, DNA based methods, antibody-based technologies and microscopy were all employed to clarify the results for the presence of nut allergens in cumin. A two-tier test system was previously reported by Black et al. [50] for the detection of adulteration in oregano. This successful process involved the use of FTIR spectroscopy followed by liquid chromatography high-resolution mass spectrometry (LC-HRMS). A similar two-tier approach such as this could perhaps be used by determining the difference in the biomarkers present in paprika and spent paprika, therefore creating a confirmatory method following screening on spectroscopy.

### 3.4. Sudan Dye

Sudan I dye was added to 100% spent and 50% spent samples. These samples were spiked at 0.1%, 0.5%, 1%, 2.5%, and 5% Sudan I dye. These samples were then predicted as unknown against the validated OPLS-DA models presented (Figure 5).

In Figure 5, it can be observed that the 50% and 100% spent paprika that had been spiked with Sudan I dye were mostly found between the Paprika and Adulterant classes (50% spent) or in a similar position of the Adulterant class (100% spent), although the separation was clearer with the NIR results. Although this method is focused on detecting the economic adulteration of paprika with spent paprika and is not focused on the detection of Sudan I dye, it is worth noting that the higher the spiking level of Sudan I dye, the further the samples moved from the Paprika class in both the NIR and FTIR models. This is important as this reduces the chance of Sudan I dye being added to cheat the method by moving the samples closer to the Paprika class.

The predicted score values on the classification list from SIMCA (Table 3) were low in comparison to the cut-off values of 0.737 (NIR) and 0.922 (FTIR), indicating adulteration with spent material. Therefore, the Sudan I dye has no effect on the outcome of the results from this test.

## 4. Conclusions

NIR and FTIR were used in conjunction with chemometrics to develop methods for the detection of spent paprika in paprika as a fraudulent bulking agent. External validation was carried out on both methods, and the NIR detected 100% of the authentic paprika samples as paprika, whereas the FTIR detected 83.33%. The NIR method detected all spiked samples in the external test set from the 50% adulteration level, whereas the FTIR detected all from the 40% adulteration level. As there is little evidence of a similar rapid screening technique for the detection of spent paprika in paprika, this method indicates potential in this area as separation was detected in the chemometric models, although low-level adulteration was not always detected. To assist with the shortfall, this method could possibly be used as part of a two-tiered system by following up questionable results with a confirmatory method. Sudan dye, a possible addition to spent material to enhance colour, was added to 100% spent paprika and 50% spent paprika to determine if it could have an effect on the test method for the detection of spent material; however, this had no effect on the result outcome.

## Figures and Tables

**Figure 1 foods-09-00944-f001:**
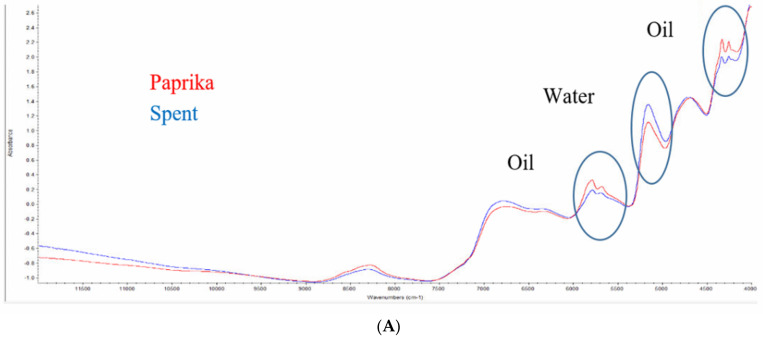
Near-infrared (NIR) (**A**) and Fourier transform infrared (FTIR) (**B**) raw spectral data of paprika and spent paprika.

**Figure 2 foods-09-00944-f002:**
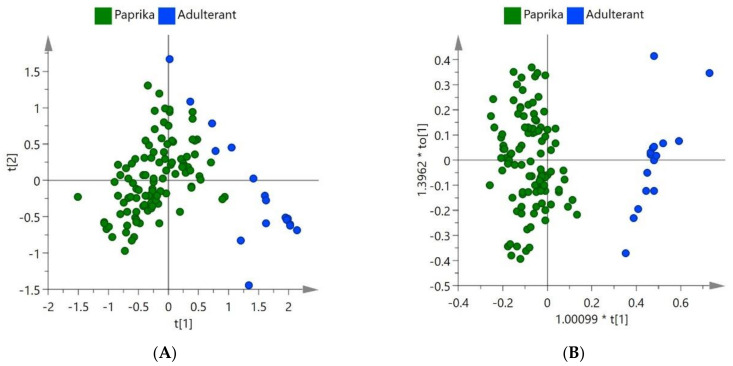
NIR (**A**) principal component analysis (PCA) (Unsupervised) and (**B**) OPLS-DA (Supervised) classification models for paprika and spent material.

**Figure 3 foods-09-00944-f003:**
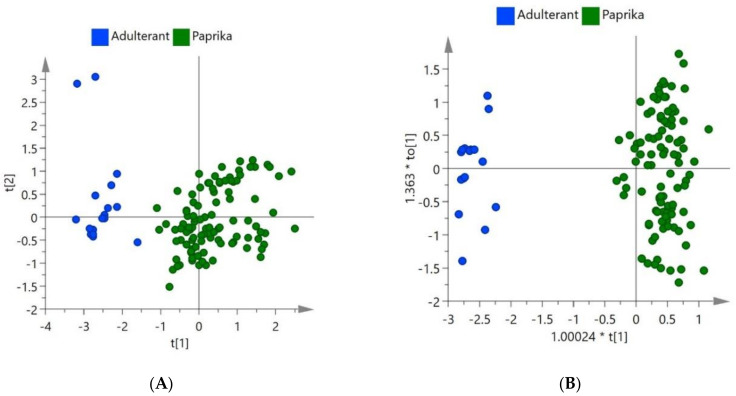
FTIR (**A**) PCA (Unsupervised) and (**B**) orthogonal partial least squares discriminant analysis (OPLS-DA) (Supervised) classification models for paprika and spent material.

**Figure 4 foods-09-00944-f004:**
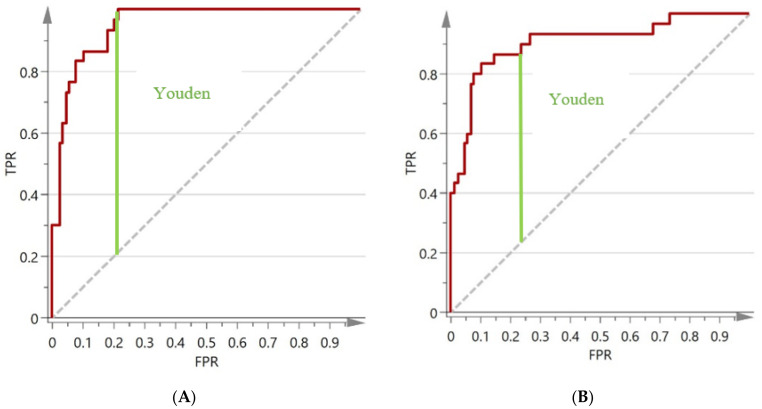
Receiver operating curves (ROC) and Youden index for NIR (**A**) and FTIR (**B**) test methods.

**Figure 5 foods-09-00944-f005:**
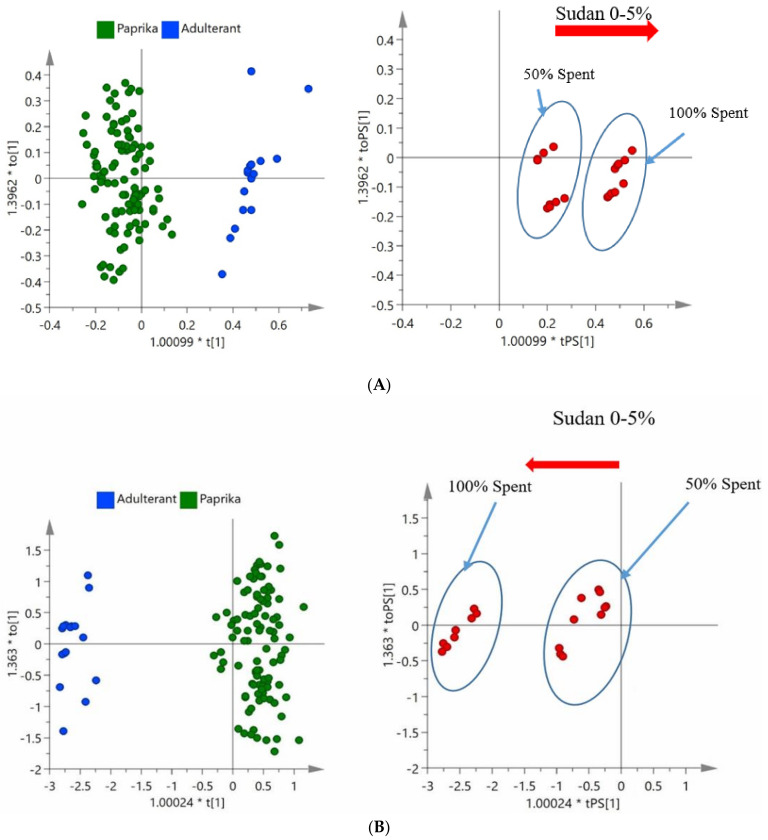
OPLS-DA models (left) and predicted 100% and 50% spent paprika spiked with Sudan I dye ranging from 0–5% (right) on NIR (**A**) and FTIR (**B**).

**Table 1 foods-09-00944-t001:** The use of spectroscopy in the detection of adulteration of paprika.

Method	Chemometrics	Adulterants	Ref.
Fourier transform infrared (FT-IR)	Principal component analysis (PCA), One class soft independent modelling class analogy (OCSIMCA)	1% Sudan I, 1% Sudan IV, 3% lead chromate, 3% lead oxide, 5% silicon dioxide, 10% polyvinyl chloride, 10% gum arabic	[4]
FT-Near-infrared (NIR)	Classical least squares (CLS)-based Advanced ID algorithm	Tomato skins, brick dust, Sudan I	[18]
NIR—Portable	Partial least squares-discriminant analysis (PLS-DA), Partial least squares regression (PLSR)	Potato starch, acacia gum, annatto	[19]
FTIR	Hybrid linear analysis (HLA)/GO	Sudan I	[20]
Raman	PLSR, PLS-DA	Sudan I	[21]
Raman hyper-spectral imaging (HSI)	Linear correlation	Sudan I and Congo Red	[22]
Surface-enhanced Raman spectroscopy (SERS)	PCA	Sudan I	[23]
Molecularly imprinted polymers-thin layer chromatography-surface enhanced Raman spectroscopy (MIP-TLC-SERS)	PCA, Linear Correlation, PLSR	Sudan I	[24]
Solution-NMR (Nuclear Magnetic Resonance), Solid-State NMR	Linear Regression	Sudan I	[25]
^1^H NMR	PLS-DA	Sudan I-IV	[26]
Synchronous fluorescence spectroscopy (SFS)	PLS-DA	Sudan I	[27]
UV-Vis	PCA, PLS-DA, PLSR	Sudan I and II	[28]
UV-Vis	PCA, PLS-DA, K-nearest neighbours (KNN)	Sudan I, Sudan I + IV blend	[29]
UV-Vis	PLS-DA, KNN, SIMCA	Sudan I, II, III and IV	[30]
UV-Vis	PCA, PLS-DA	Sudan I and IV	[31]

**Table 2 foods-09-00944-t002:** Correct classification rate of external validation set for NIR and FTIR paprika adulteration test methods.

	NIR Correct Classification %	FTIR Correct Classification %
100% Paprika	100%	83.3%
10% Spent	20%	50%
20% Spent	30%	70%
30% Spent	70%	90%
40% Spent	90%	100%
50% Spent	100%	100%
60% Spent	100%	100%
70% Spent	100%	100%
80% Spent	100%	100%
90% Spent	100%	100%

**Table 3 foods-09-00944-t003:** The averaged predicted score values of 50% and 100% spent samples following spiking with Sudan I dye.

% Spent	% Sudan 1	NIR	FTIR	% Spent	% Sudan 1	NIR	FTIR
Cut-off		0.737	0.922	Cut-off		0.737	0.922
50%	0.10%	0.587	0.769	100%	0.10%	0.152	0.112
50%	0.50%	0.577	0.769	100%	0.50%	0.140	0.115
50%	1%	0.578	0.715	100%	1%	0.129	0.075
50%	2.50%	0.538	0.602	100%	2.50%	0.094	−0.047
50%	5%	0.480	0.574	100%	5%	0.043	−0.109

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
