# Peer review of "The Detection of Substitution Adulteration of Paprika with Spent Paprika by the Application of Molecular Spectroscopy Tools"

_foods, 2020, doi:10.3390/foods9070944_

Round 1
Reviewer 1 Report
The manuscript describes the application of unsupervised (PCA) and supervised (OPLS-DA) chemometric discrimination tools to NIR and FTIR measurements on paprika samples in order to detect adulterations, mainly by spent paprika. The number of samples considered in the study is really high (159) and there is a carefully planned experimental design and a critical evaluation of the results.
The performance of both NIR- and FTIR-based methodologies is promising, especially considering the speed of the analysis, with a very simple data treatment and a fast measurement. The main drawback, however, is that these methods need a quite high proportion of adulterant to detect the fraud, which demands a further improvement of the methods or a combination with other techniques. A good point is that the adulteration with other substances like Sudan I dye does not seem to affect the performance of the IR methods.
- In the Introduction, when the authors talk about different methods of paprika authentication, they should include some chromatographic approaches based on the obtention of characteristic fingerprints of relevant constituents of the samples as, for instance, phenolic substances. You can see for instance: https://doi.org/10.1016/j.talanta.2018.06.085 and https://doi.org/10.1016/j.lwt.2020.109153
- Have the authors though about combining NIR and FTIR spectra in augmented matrices to improve the performance of the method?
- In Figure 1A, I suppose 'Oi' must be 'Oil'.
- In R2 and Q2 parameters I would change '2' to superscript format.
- In Fig,5B, right, 'Spe' should be 'Spent'.
Author Response
The following changes have been made to the manuscript based on the constructive comments of the reviewers and the authors wish to thank them for their help in improving the manuscript.
Reviewer 1
- In the Introduction, when the authors talk about different methods of paprika authentication, they should include some chromatographic approaches based on the obtention of characteristic fingerprints of relevant constituents of the samples as, for instance, phenolic substances. You can see for instance: https://doi.org/10.1016/j.talanta.2018.06.085 and https://doi.org/10.1016/j.lwt.2020.109153
- Both of these methods have been included in the introduction to show the benefits of chromatographic approaches in paprika adulteration detection in lines 58-61.
- Have the authors though about combining NIR and FTIR spectra in augmented matrices to improve the performance of the method?
- The authors decided to carry out a comparison of the NIR and FTIR methods to see if one performed better than the other. As there is reduced sample preparation with the NIR method, if this method gives similar results, it may provide a faster approach for analysis. This may also give an indication of future handheld spectroscopy possibilities, an area the authors are interested in. However, although not carried out in this study, there is a great likelihood of improved performance in combining the data and may be looked into in the future.
- In Figure 1A, I suppose 'Oi' must be 'Oil'.
- The word on figure 1A has been corrected to ‘Oil’.
- In R2 and Q2 parameters I would change '2' to superscript format.
- The R2 and Q2 values have been changed to R2 and Q2 throughout the manuscript.
- In Fig,5B, right, 'Spe' should be 'Spent'.
- The word Spe has been corrected to ‘Spent’.
Reviewer 2 Report
The manuscript entitled: “The detection of substitution adulteration of paprika with spent paprika by the application of molecular spectroscopy tools” aims at detecting adulterated paprika by means of MIR and NIR spectroscopy coupled with chemometrics (OPLS-DA).
The study is interesting, and it is definitely of interest for the readers of Foods.
The manuscript is quite clear. The rationale behind the research is properly reported and the state of the art is well described. Nevertheless, I think there is still room for improvements; consequently, I would like to provide some comments/suggestions to the authors.
Major Comments
1) Facing the different classification problems, it is not always completely clear how many samples per class you have. Please, try to clarify this in the manuscript.
2) The data set described is what is called a multi-block data set, because the same samples have been analyzed by two NIR and MIR. So far, it is quite well-known it is better to handle this kind of data by means of multi-block (data fusion) techniques. Why did you not take it into considerations?
I am aware of the possibility of freely downloading multi-block functions for matlab online, for instance, here:
https://www.chem.uniroma1.it/romechemometrics/research/algorithms/so-pls/
you can download two multi-block classifiers: SO-PLS and SO-CovSel which would be suitable for your work.
In case you currently do not have access to matlab, I think you can have a free demo for a short period of time.
In case you are not familiar with matlab, I think you can find multi-block approaches running in R.
I do not know if the software you used for calculations allows multi-block analysis. In case it does, you can use whatever data fusion method is provided.
Eventually, in case you are not interested in including multi-block calculations in the present manuscript, please, motivate your choice in the answer to the reviewers.
3) Section 2.3. Line 138. Spectra were collected in reflectance mode. Have you transformed them in pseudo-absorbance prior to the creation of the classification model?
4)Discussing the outcomes of the OPLS-DA analysis you provide the R2. Due to the fact you are applying a PLS-based classifier, I understand you have this information, but this is usually used to evaluate the fit in regression; in classification you have more appropriate indices of the goodness of your model. Why didn’t you report something more naturally related to classification? Classification rates, or the classification errors, for instance. Additionally, in calibration, it is not mentioned if R2 is validated or not. Moreover, please check how R2 is written, sometimes R2 is used instead of R2 .
Minor comments
Derivatives were used as a pretreatment. Did you use the Savitzky-Golay approach? In case you did, please, provide the parameters you used (points of the window and degree of the polynomial). In case you use another approach, please, mention which one.
Author Response
The following changes have been made to the manuscript based on the constructive comments of the reviewers and the authors wish to thank them for their help in improving the manuscript.
Reviewer 2
Major Comments
- Facing the different classification problems, it is not always completely clear how many samples per class you have. Please, try to clarify this in the manuscript.
The total number of samples used in the study was 140 authentic paprika and 19 spent paprika.
Models: The chemometric models were made up of 17 spent paprika samples and 104 authentic paprika samples.
External validation set: Two spent paprika samples were used for the spiking in the external validation set. Thirty authentic paprika samples were used in the validation set and 6 were used for spiking. Five out of the 6 spent samples were used to spike each of the two spent materials. These values have now been included in lines 196-197 and 213-216.
- The data set described is what is called a multi-block data set, because the same samples have been analyzed by two NIR and MIR. So far, it is quite well-known it is better to handle this kind of data by means of multi-block (data fusion) techniques. Why did you not take it into considerations? I am aware of the possibility of freely downloading multi-block functions for matlab online, for instance, here: https://www.chem.uniroma1.it/romechemometrics/research/algorithms/so-pls/
you can download two multi-block classifiers: SO-PLS and SO-CovSel which would be suitable for your work. In case you currently do not have access to matlab, I think you can have a free demo for a short period of time. In case you are not familiar with matlab, I think you can find multi-block approaches running in R. I do not know if the software you used for calculations allows multi-block analysis. In case it does, you can use whatever data fusion method is provided. Eventually, in case you are not interested in including multi-block calculations in the present manuscript, please, motivate your choice in the answer to the reviewers.
The authors decided to carry out a comparison of NIR and FTIR methods to see if one performed better than the other. As there is reduced sample preparation with the NIR method, if this method gives similar results, it may provide a faster approach for analysis. This comparison can also give an indication into future possibilities with handheld spectroscopy equipment which is an area the authors are interested in. However, although not carried out in this study, there is a great likelihood of improved performance in combining the data and may be looked into in the future, and we agree that investigation into this is important in the future. Also, the software used for model development was SIMCA 15, and this does not allow for data fusion.
- Section 2.3. Line 138. Spectra were collected in reflectance mode. Have you transformed them in pseudo-absorbance prior to the creation of the classification model?
Line 165 has been updated to say “Data was collected in reflectance mode, with spectral data output measured in absorbance units”.
- Discussing the outcomes of the OPLS-DA analysis you provide the R2. Due to the fact you are applying a PLS-based classifier, I understand you have this information, but this is usually used to evaluate the fit in regression; in classification you have more appropriate indices of the goodness of your model. Why didn’t you report something more naturally related to classification? Classification rates, or the classification errors, for instance. Additionally, in calibration, it is not mentioned if R2is validated or not. Moreover, please check how R2 is written, sometimes R2 is used instead of R2 .
The R2 values have now been changed to R2 in the manuscript. In SIMCA 15, the goodness of fit for a classification model is represented by R2 values. The external validation set, with known authentic samples and known adulteration levels provided the ability of the models to correctly classify. The Q2 value was also used to determine the ability of the model to predict.
Minor comments
- Derivatives were used as a pretreatment. Did you use the Savitzky-Golay approach? In case you did, please, provide the parameters you used (points of the window and degree of the polynomial). In case you use another approach, please, mention which one.
Savitzky-Golay was used with 15 points and a quadratic polynomial order. This is now included in the article in section 2.5.
Reviewer 3 Report
In Introduction some additional lines on nutraceutical values of foods should be added such as:
Santini A, Cammarata SM, Capone G, et al. Nutraceuticals: opening the debate for a regulatory framework. Br J Clin Pharmacol. 2018;84(4):659-672. doi:10.1111/bcp.13496
The advantages of chemometrics approach should be better explained in Introduction and some lines on several types of chemometric application on food and thereof as well as complex food matricies should be mentioned such as:
Durazzo A., Kiefer J., Lucarini M., Camilli E., Marconi S., Gabrielli P., Aguzzi A., Gambelli L., Lisciani S., and Marletta L. Qualitative analysis of Traditional Italian Dishes: FTIR approach. Sustainability, 2018, 10, 4112; doi:10.3390/su10114112
Updated references should be added to Table 1.
Figure 1 should be better discussed in the text with major argumentations
The discussion of lines 224-232 should be enlarged.
Major details should be given in lines 240-245.
Lines 261-265 should be clarified.
Author Response
The following changes have been made to the manuscript based on the constructive comments of the reviewers and the authors wish to thank them for their help in improving the manuscript.
Reviewer 3
- In Introduction some additional lines on nutraceutical values of foods should be added such as: Santini A, Cammarata SM, Capone G, et al. Nutraceuticals: opening the debate for a regulatory framework. Br J Clin Pharmacol. 2018;84(4):659-672. doi:10.1111/bcp.13496
The authors do not feel that the addition of information on nutraceutical value of paprika is appropriate for this paper. This paper focuses on the adulteration issues with paprika, and not on the value or benefits of the food.
- The advantages of chemometrics approach should be better explained in Introduction and some lines on several types of chemometric application on food and thereof as well as complex food matricies should be mentioned such as: Durazzo A., Kiefer J., Lucarini M., Camilli E., Marconi S., Gabrielli P., Aguzzi A., Gambelli L., Lisciani S., and Marletta L. Qualitative analysis of Traditional Italian Dishes: FTIR approach. Sustainability, 2018, 10, 4112; doi:10.3390/su10114112
The authors have included a sentence to further clarify the advantages of chemometrics, as well as including examples of its use in the detection of adulteration in other herbs and spices. The authors wanted to keep a focus on the detection of adulteration in herbs and spices in this manuscript. See lines 70-73.
- Updated references should be added to Table 1.
The references have been updated in Table 1 following the addition of new references.
- Figure 1 should be better discussed in the text with major argumentations
Further discussion has been included in this section to support the process of method development in lines 243-259.
- The discussion of lines 224-232 should be enlarged.
A few sentences on the PCA and OPLS-DA models have been included in this section to further explain the process and reasons behind the model development in lines 266-275.
- Major details should be given in lines 240-245.
Further details have been added to the lines 284-293. This includes the reason for randomization and the sample numbers used at each step.
- Lines 261-265 should be clarified.
Further clarification has been added to this paragraph indicating the purpose of the cut-off value in lines 307-311.
Round 2
Reviewer 2 Report
The authors replied to the reviewer's comments without providing substantial changes in the manuscript.
How the external validation sample are organized is not clear. In particular, the authors said:
"External validation set: Two spent paprika samples were used for the spiking in the external validation set. Thirty authentic paprika samples were used in the validation set and 6 were used for spiking. Five out of the 6 spent samples were used to spike each of the two spent materials.".
Please clarify this.
Author Response
The authors would like to thank the reviewer for their comments and have amended the manuscript foods- 848700, ‘The detection of substitution adulteration of paprika with spent paprika by the application of molecular spectroscopy tools’ as requested. Please find the amendment below.
Comments and Suggestions for Authors
The authors replied to the reviewer's comments without providing substantial changes in the manuscript.
How the external validation sample are organized is not clear. In particular, the authors said:
"External validation set: Two spent paprika samples were used for the spiking in the external validation set. Thirty authentic paprika samples were used in the validation set and 6 were used for spiking. Five out of the 6 spent samples were used to spike each of the two spent materials.".
Please clarify this.
These sentences have now been added to the manuscript to clarify the external validation set.
‘All samples chosen for external validation were removed from the chemometric model set. The external validation set was made up of authentic (typical) paprika samples (30) and spiked samples (atypical) (90). To carry out the spiking, two spent paprika were used. Each spent paprika was used to spike five authentic paprika samples at 10-90% levels, therefore resulting in 90 spiked samples in total. The five authentic paprika samples chosen for spiking were selected out of a range of six samples’. This is outlined in lines 213-218.